# Evaluation of Sodium Relaxation Times and Concentrations in the Achilles Tendon Using MRI

**DOI:** 10.3390/ijms231810890

**Published:** 2022-09-17

**Authors:** Benedikt Kamp, Miriam Frenken, Lena Klein-Schmeink, Armin M. Nagel, Lena M. Wilms, Karl Ludger Radke, Styliani Tsiami, Philipp Sewerin, Xenofon Baraliakos, Gerald Antoch, Daniel B. Abrar, Hans-Jörg Wittsack, Anja Müller-Lutz

**Affiliations:** 1Department of Diagnostic and Interventional Radiology, Medical Faculty, University Dusseldorf, D-40225 Dusseldorf, Germany; 2Institute of Radiology, University Hospital Erlangen, Friedrich-Alexander-Universität Erlangen-Nürnberg (FAU), D-91054 Erlangen, Germany; 3German Cancer Research Center (DKFZ), Division of Medical Physics in Radiology, D-69120 Heidelberg, Germany; 4Rheumazentrum Ruhrgebiet, Ruhr-University Bochum, Claudiusstr. 45, D-44649 Herne, Germany; 5Department and Hiller Research Unit of Rheumatology, Heinrich Heine University Düsseldorf, UKD, Moorenstrasse 5, D-40225 Düsseldorf, Germany

**Keywords:** sodium MRI, ^23^Na MRI, sodium relaxation time, sodium concentration, Achilles tendon, tendon, proteoglycan, glycosaminoglycan, collagen

## Abstract

Sodium magnetic resonance imaging (MRI) can be used to evaluate the change in the proteoglycan content in Achilles tendons (ATs) of patients with different AT pathologies by measuring the ^23^Na signal-to-noise ratio (SNR). As ^23^Na SNR alone is difficult to compare between different studies, because of the high influence of hardware configurations and sequence settings on the SNR, we further set out to measure the apparent tissue sodium content (aTSC) in the AT as a better comparable parameter. Ten healthy controls and one patient with tendinopathy in the AT were examined using a clinical 3 Tesla (T) MRI scanner in conjunction with a dual tuned ^1^H/^23^Na surface coil to measure ^23^Na SNR and aTSC in their ATs. ^23^Na T_1_ and T_2_* of the AT were also measured for three controls to correct for different relaxation behavior. The results were as follows: ^23^Na SNR = 11.7 ± 2.2, aTSC = 82.2 ± 13.9 mM, ^23^Na T_1_ = 20.4 ± 2.4 ms, ^23^Na T_2s_* = 1.4 ± 0.4 ms, and ^23^Na T_2l_* = 13.9 ± 0.8 ms for the whole AT of healthy controls with significant regional differences. These are the first reported aTSCs and ^23^Na relaxation times for the AT using sodium MRI and may serve for future comparability in different studies regarding examinations of diseased ATs with sodium MRI.

## 1. Introduction

The Achilles tendon (AT) connects the gastrocnemius and soleus muscle to the calcaneus and has to endure loads up to 12.5 times of the subject’s body weight [1]. The main component of the healthy AT is collagen (70% of dry weight), which in turn is composed predominantly of type I collagen fibers (95%) and to a lesser extent of other collagen types [2]. Proteoglycans (PG) with glycosaminoglycan (GAG) side chains of varying types contribute another 1.2% to the dry weight of the tendon [3]. The AT receives its blood supply from the musculotendinous and the osteotendinous junction, as well as the surrounding connective tissue [4]. However, vascularity in total is relatively poor, which may lead to inadequate tissue repair mechanisms following trauma [5]. The middle portion of the AT—which is also the thinnest part—has an even more reduced blood supply [5]. The combination of these two factors might be the reason why AT ruptures are most common in this region [6].

Tendinopathy is often characterized as a failed healing response, mainly as a consequence of repetitively overloading the tendon [7,8,9]. Symptoms include pain, reduced performance, and swelling in and around the tendon [2,7,8]. Clinical standard magnetic resonance imaging (MRI) is used to obtain morphological images of the AT and the surrounding structures for a more accurate diagnosis [7]. However, early stages of tendinopathy are not accompanied by morphological changes, but initially by biochemical alterations, which are not visible in standard MRI [3]. In the pathological tendon, both the water content and the fraction of collagen type III in relation to collagen type I increase [10]. Furthermore, the PG and GAG content increase and the types of present PGs and GAGs change [9,11,12]. A variety of different quantitative MRI techniques have been proposed to detect these biochemical changes.

As pathologies such as tendinopathy or tendon rupture are characterized by less orderly arranged collagen structures and increased water content, they can be measured using ^1^H T_2_ and ultra short echo time (UTE) T_2_* mapping [3,13,14,15,16]. Other techniques such as chemical exchange saturation transfer (CEST), T_1_ρ, and ^23^Na imaging indicate promising results to assess biochemical components such as PG and GAG content [17,18,19,20]. Previous studies have linked ^23^Na signal-to-noise ratio (SNR) values to GAG content in the AT, first in cadaver specimens using histological analysis and later also in vivo [3,21,22].

While ^23^Na SNR values can already be a good marker for GAG content, ^23^Na imaging is most commonly used for the calculation of tissue sodium concentration (TSC) [23,24,25,26,27,28]. The calculation of TSC provides a standardized parameter that already includes a number of correction methods and allows comparison of studies with different hardware and imaging sequences. For TSC calculation, phantoms with known ^23^Na concentrations, which have similar ^23^Na relaxation properties as the examined tissue, should be placed in the imaging field of view (FOV) [29,30]. Furthermore, in most cases an additional correction for different relaxation times of the phantoms and the tissue is needed [27,31,32]. In many cases it is also advisable to correct for the influence of different ^23^Na imaging resolutions on the calculated TSC using a partial volume correction method [24,31]. ^23^Na MRI has inherently a low imaging resolution, which can lead to sometimes severe partial volume effects and consequent miscalculation of TSC [24,32]. Moreover, corrections for coil sensitivity are often necessary, especially when using a surface coil [24,25,31]. However, it has not yet been examined if the total sodium content in tendons is MR visible with the chosen acquisition parameters and how the residual quadrupolar interaction of ^23^Na ions in tendons influences the signal intensity. Thus, we will use the term apparent tissue sodium concentration (aTSC) instead of TSC, as suggested by Stobbe and Beaulieu [33,34].

We aim to determine ^23^Na parameters for the AT using a clinical 3 Tesla (T) MR scanner by first measuring ^23^Na T_1_ and T_2_* relaxation times. Then we use the calculated relaxation times and apply the correction methods mentioned above to determine aTSC. Furthermore, we measure ^23^Na SNR and ^1^H T_2_* to compare the values with those of previous studies, to provide a reference point for future ^23^Na AT studies particularly using aTSC as a more standardized parameter. We hypothesize that it is possible to determine aTSC and ^23^Na relaxation times in vivo in the human AT and to observe to a certain extent similar trends between aTSC, ^23^Na SNR and ^1^H T_2_* measurements.

## 2. Results

^23^Na relaxation times were successfully determined for all three studied AT regions of interests (ROIs), hence tendon insertion point into the calcaneus (INS), middle portion of the tendon (MID) and myotendinous junction (MTJ), and are summarized in detail in Table 1. ^23^Na T_1_ increased slightly with the distance to the insertion point into the calcaneus from 18.4 ± 2.7 ms in the INS ROI to 20.4 ± 2.4 ms in the MTJ ROI. ^23^Na T_2_* values were similar across the three different regions with ^23^Na T_2s_* = 1.4 ± 0.4 ms, ^23^Na T_2l_* ≈ 14 ± 1 ms and p_s_ ≈ 32 ± 3%. The mean ^23^Na relaxation times across all phantom ROIs and healthy control measurements were determined to be ^23^Na T_1_ = 38.5 ± 3.8 ms, ^23^Na T_2s_* = 6.0 ± 0.5 ms and ^23^Na T_2l_* = 13.0 ± 1.6 ms. Exemplary fits for ^23^Na T_1_ fitting of one healthy control are displayed in Figure 1 and Figure 2 shows exemplary fits for ^23^Na T_2_* fitting of one healthy control.

aTSC, ^23^Na SNR and ^1^H T_2_* values were also successfully determined for all healthy controls and the patient with Achilles tendinopathy. The resulting values are displayed in Table 2 and further illustrated overlaying colormaps of aTSC, ^23^Na SNR and ^1^H T_2_* values onto morphological ^1^H images in Figure 3. In Figure 4, clinical sagittal images of the same participants as in Figure 3 are displayed for anatomical reference. The three ROIs of the ATs of the healthy controls were compared with respect to their biochemical properties using appropriate statistical tests. The *p*-values are indicated in Table 3 and are visualized in Figure 5 using boxplots.

For the healthy volunteers, with increasing distance of the AT to the calcaneal insertion, the mean aTSC values decreased (INS: aTSC = 112.9 ± 21.1 mM vs. MID: aTSC = 77.3 ± 13.3 mM vs. MTJ: aTSC = 55.3 ± 13.3 mM), with significant differences in aTSC between all ROIs (INS-MID: *p* = 0.015, INS-MTJ: *p* = 0.015, MID-MTJ: *p* = 0.015). The relative standard deviation (SD) of aTSC within the ROIs of each control was not significantly different at approximately 28 ± 5% as mean SD over all ROIs and healthy controls. Compared to healthy controls, the patient indicated comparable mean aTSC values for INS (89.3 ± 37.7 mM) and MTJ (51.3 ± 19.9 mM), while the mean aTSC value for MID was slightly higher (90.6 ± 35.2 mM). The trend of decreasing aTSC values with higher distance to INS, as seen in the healthy controls, could not be observed in the patient. While the relative SDs of the patient were also similar for all ROIs at approximately 39 ± 2%, the relative SDs of the patient were higher than the SDs of the healthy controls.

For the ^23^Na SNR values, similar observations could be made. For the healthy controls, mean ^23^Na SNR values also decreased with increasing distance to the calcaneal insertion from INS ^23^Na SNR = 14.2 ± 2.8 to MID ^23^Na SNR = 10.7 ± 2.2 to MTJ ^23^Na SNR = 9.5 ± 2.1. Moreover, significant differences could be observed between these ROIs (INS-MID: *p* = 0.015, INS-MTJ: *p* = 0.015, MID-MTJ: *p* = 0.015). For the relative SDs of ^23^Na SNR values, significant differences could only be observed for INS (24.3 ± 3.1%) vs. MTJ (20.2 ± 1.6%) with *p* = 0.028. While the patients ^23^Na SNR values (INS: 11.8 ± 3.7, MID: 12.3 ± 3.5 and MTJ: 9.5 ± 2.2) were again comparable to those of the healthy controls, the trend towards decreasing ^23^Na SNR with increasing distance to INS could not be observed in the patient. The relative ^23^Na SNR SDs of the patient were slightly higher for INS (31.2%) and MID (28.8%) compared to the healthy controls, while the value for MTJ (23.2%) was of similar size.

Mean ^1^H T_2_* values for healthy controls slightly increased with higher distance to INS (INS: 1.9 ± 0.1 ms vs. MID: 2.0 ± 0.3 ms vs. MTJ: 2.3 ± 0.6 ms). However, only the difference between MID and MTJ was significant (*p* = 0.038). The ^1^H T_2_* SDs for MID and MTJ were similar at approximately 33 ± 4%, while the SD for INS was slightly lower at 26 ± 5%, yet non-significant (*p* = 0.061). In comparison, the patient exhibited higher mean ^1^H T_2_* values, especially in MID (3.7 ± 1.3 ms) and MTJ (3.5 ± 1.0 ms). The SDs of the ^1^H T_2_* values within the ROIs were comparable to the values of the healthy controls.

## 3. Discussion

The most important finding of our study is that, for the first time, successful measurements of ^23^Na relaxation times and aTSC in the ATs were obtained using MRI. These parameters are comparable between different MRI sequences and hardware configurations and open new possibilities to study ATs biochemically and to improve diagnostic capabilities for different AT diseases. aTSC is especially useful for assessing GAG content, which changes with different tendon pathologies [3,9,11,12]. In addition, our study was performed with a field strength of 3T, which is cheaper and more widely used in clinics than 7T MRI scanners, reducing the barrier towards clinical application.

Values for ^23^Na relaxation times in the AT using MRI were not previously published to the best of our knowledge, which inhibits the direct comparison of our values to the literature. However, our estimated relaxation times for the whole AT (^23^Na T_1_ = 20.4 ± 2.4 ms; ^23^Na T_2s_* = 1.4 ± 0.4 ms; ^23^Na T_2l_* = 13.9 ± 0.8 ms; p_s_ = 31.6 ± 2.6%) are close but in general slightly higher than previously published relaxation time values for articular cartilage at 3T (^23^Na T_1_ = 14.5 ± 0.7 ms; ^23^Na T_2s_* = 0.4 ± 0.1 ms; ^23^Na T_2l_* = 12.6 ± 0.7; p_s_ = 34 ± 5%) [32]. The similarity might be due to a similar collagen content in AT (70% of dry weight, mainly collagen type I) and articular cartilage (60% of dry weight, mainly collagen type II). Higher collagen content in combination with more orderly alignment of the fibers is considered to shorten the ^1^H T_2_* times of the AT [15]. If this reasoning were to be applied for ^23^Na relaxation times, it could be deducted, tendon ^23^Na relaxation times should be slightly lower than those of cartilage. However, other factors may also influence differences in relaxation behavior between cartilage and the AT, such as different collagen types, more synovial fluid in close proximity to cartilage and different PG types and content [35]. Because of quantum mechanical properties, p_s_ should theoretically be 60% in a single pool of ^23^Na ions [36]. Our results for the AT deviate from this; however, tissue can rarely be considered as a single pool of ^23^Na ions, and for articular cartilage similar deviations from the theoretical 60% have been published [31,32,36,37]. In contrast, our agarose phantoms represent a much more controlled environment, in which ^23^Na ions might be considered as a single pool, which is reflected by our result of p_s_ being close to 60%.

The general decrease in ^23^Na SNR from INS to MTJ in the AT was previously described in two different studies by Juras et al. conducted at a higher field strength of 7T [3,21]. In the AT of cadaver specimens, the mean ^23^Na SNR and its interquartile range (IQR) was reported as 9.6 (IQR: 8.0–14.1) for the whole AT, which is within SD range of our results for healthy controls [3]. In the AT of healthy controls a ^23^Na SNR value of 4.9 ± 2.1 was reported for the whole tendon, which is considerably lower than our values [21]. This illustrates the challenges of directly comparing ^23^Na SNR values, as the absolute SNR values are dependent upon many factors such as sequence settings (e.g., TE, TR, averages), different MRI scanners and different coils. However, the trend towards lower SNR with increasing distance to INS is comparable and observable in both Juras’ and our study [21].

There is no published study investigating aTSC in ATs or at least in other tendons or ligaments to the best of our knowledge, so a comparison of aTSC with the literature values is difficult. Articular cartilage has approximately four times more GAG by weight compared to the AT (5% vs. 1.2%) and the TSC of healthy articular cartilage is considered to be between 220 mM and 270 mM, which is approximately three times higher than our result of 82 mM for the whole AT [3,26,38]. This ratio seems to be reasonable, considering different ^23^Na relaxation times, collagen content, imaging techniques and corrections with varying precisions based on the aforementioned also influence the calculated aTSCs for the different tissue types. Our result of approximately 82 mM for the aTSC in the whole AT is considerably lower than the expected sodium concentration of 140 mM in the extracellular space [39]. The AT is mainly composed of a collagen-rich extracellular matrix, which would indicate the aTSC values should be closer to the extracellular sodium concentration of 140 mM [40]. However, sodium concentration measurements using MRI have previously been reported to obtain lower values than, for example, chemical analysis, which might also be the case for our aTSC values for the AT [41]. Reasons could be an underestimation of the influence of partial volume effects in our correction or spatial deviations of the B**_0_** and B**_1_** fields between the positions of the agarose reference phantoms and the AT. This is another reason why our resulting sodium concentrations should be referred to as “apparent tissue sodium concentration”, because validation of the accuracy of measurement of sodium concentration in the AT by MRI would need to be investigated in a future study by comparison with, for example, chemical analysis.

The determined values for ^1^H T_2_* in healthy controls are similar to the results of other studies. Chen et al. compared ^1^H T_2_* values between healthy controls and patients suffering from psoriatic arthritis with inflammation of the AT [42]. They measured 1.33 ± 0.11 ms for the enthesis of the AT and 0.88 ± 0.02 ms for the rest of the AT of healthy controls as well as 2.66 ± 0.61 ms for the enthesis of the AT and 2.22 ± 0.58 ms for the rest of the AT of their patients [42]. In line with our measurements, they also reported elongated ^1^H T_2_* values in patients with diseased ATs. In general, their ^1^H T_2_* values are slightly shorter than ours, which might be due to them using different TEs in their study compared to ours (0.03 ms, 0.6 ms, 4.4 ms, 8.8ms), especially the first two TEs in close succession below 1 ms might lead to shorter relaxation times from the fitting process. In a study by Filho et al., where ten different TEs (0.1 ms, 0.2 ms, 0.3 ms, 0.5 ms, 0.75 ms, 1 ms, 2 ms, 4 ms, 8 ms, 15 ms) were used to determine ^1^H T_2_* of the AT in six cadaveric specimens, their result for ^1^H T_2_* of the whole AT was 2.18 ± 0.30 ms, which is closer to our results [43].

The differences between the ROIs INS, MID and MTJ for aTSC and ^23^Na SNR values were to be expected and are likely caused by biochemical differences between the different tendon sections. The insertional part of the AT has a higher GAG content than other segments and the collagen matrix is comprised of more collagen type II [44]. The reason is, that in the insertional region of the AT not only tensional but also compressive forces act on the AT, which leads to fibrocartilaginous structures that are similar in biochemical composition to articular cartilage [44,45]. Correspondingly, we observed significantly higher aTSC and ^23^Na SNR in INS compared to other AT regions. Our mean values for ^1^H T_2_* are slightly lower in INS compared to the other regions, which might indicate the difference in collagen composition, but this difference was not statistically significant for our data. Our morphologic images of the patient showed signs of tendinopathy in the area of the medial tendon, namely focal thickening and increased fluid accumulation between the tendon and skin. Our measured aTSC and ^23^Na SNR for the patient were also only increased compared to our healthy controls for the MID part of the tendon. This is a different finding compared to Juras et al., who reported increased ^23^Na SNR for the whole tendon in cases of tendinopathy, hinting towards increased GAG content in the whole tendon instead of only in the MID part [21]. However, this deviating result has not yet been statistically validated and would need to be verified by further measurements in patients.

A number of limitations of our study need to be considered. ^23^Na images inherently have a low resolution, which makes them prone to partial volume effects. While in our study a correction was applied to mitigate the resulting underestimation of especially aTSC, our correction method cannot revert the influence of surrounding tissue with ^23^Na signal and its different relaxation properties. Very close to the AT is the skin, which has a reported TSC of approximately 30–60 mM and different relaxation properties compared to the AT, which might skew our aTSC results for the AT [28]. The diseased ATs of patients could also have different ^23^Na relaxation times, as is the case for degraded articular cartilage compared with healthy controls [46]. Determining the ^23^Na relaxation times in patients may lead to a more accurate estimation of aTSC in their ATs, but imaging times and protocol numbers would have to be reduced, preferably below 60 min and only one protocol, as they are not tolerable for patients in their current state. Another factor, especially in patients, can be the ^23^Na signal of accumulating fluid. For TSC measurements in articular cartilage many different studies have been conducted to reduce the influence of synovial fluid on the differentiability of healthy control groups and patients [32,47,48,49]. Achilles tendinopathy patients are also expected to have increased fluid accumulation near the AT, which may hamper comparability between healthy and diseased subjects [7,15]. However, while for articular cartilage an inversion pulse to suppress fluid signal is feasible, the same may be very difficult to apply for measuring the AT. The AT inherently has low ^23^Na signal and the inversion pulse would further lower the ^23^Na SNR of the tendon. We instead used as high of a resolution as feasible with our equipment, but even smaller voxel sizes than our 2 mm × 2 mm × 2 mm would be helpful. In our case, more projections or averages for more ^23^Na signal and consequently higher imaging resolutions would have led to too long examination times. We were already approaching a measurement time of one hour, especially for the relaxation time measurements of the ATs. Other approaches towards accelerating ^23^Na imaging would have to be used in conjunction with more projections and signal averages to obtain more ^23^Na signal in the tendon. This would allow to further increase imaging resolution while keeping examination duration reasonable. Examples could be MR Fingerprinting and deep learning (DL) supported fitting algorithms for faster ^23^Na relaxation time measurements and compressed sensing techniques for overall faster ^23^Na image acquisition for both aTSC and relaxation time calculations [50,51,52,53].

Our results have limited capability for comparing the ^23^Na values of the healthy control groups with patients, because we only measured one patient. For a reliable comparison, a much higher number of patients would have to be measured in a future study. Furthermore, age matching of the two groups could be necessary, as vascularity, collagen type I, PG and GAG content have been shown to change with the age of the subject, which would influence the expected aTSC of the tendon in the healthy control [2,4,54]. However, our study shows the feasibility of estimating aTSC values in the diseased AT of patients in reasonable examination times.

## 4. Materials and Methods

Three separate MRI protocols were conducted. In protocols 1 and 2, the same three healthy controls were measured for ^23^Na T_1_ and ^23^Na T_2_* determination, respectively. Protocol 3 was used to estimate the aTSC and calculate ^1^H T_2_* in the AT of all participants, including the three controls measured with protocol 1 and 2, and to examine their AT in line with clinical standard procedures.

### 4.1. Study Population

Ten healthy controls (six females, four males, mean age 25.4 ± 0.9 years) and one patient with tendinopathy of the AT (female, 55 years, established by patient history and clinical MRI) participated in this study. All participants underwent imaging of their right AT with protocol 3 and three of the healthy controls (one female, two males, mean age 25.3 ± 0.5 years) were further examined with protocol 1 and 2. Participants were excluded from the healthy control group if they reported a history of acute or chronic pain in the region of the right AT or ankle. They were also excluded if pathologies of the AT were previously reported by the controls themselves or detected by the radiologist (M.F., 6 years of experience in musculoskeletal imaging) during the evaluation of the clinical MRI.

Written informed consent was obtained from all participants and the study was approved by the local ethics committee (Ethics Committee, Medical Faculty of the Heinrich-Heine-University Düsseldorf, healthy controls: study number 4733R, patient: study number 3980).

### 4.2. MRI

All imaging was conducted with a 3T MRI scanner (Siemens MAGNETOM Prisma, Siemens Healthineers, Erlangen, Germany). ^23^Na images were acquired using a dual-tuned ^23^Na/^1^H surface coil (RAPID Biomedical GmbH, Rimpar, Germany) with an 11 cm circular ^23^Na resonator and an 18 cm × 24 cm rectangular ^1^H resonator. For measurements with the dual-tuned coil, participants were positioned supine, head first, with the coil placed under the center of the right AT. All imaging with the dual-tuned coil was conducted using a density-adapted 3D radial (DA-3D-RAD) sequence [55]. For imaging following the clinical standard procedures the participants were positioned feet first and supine and their right foot was placed into a ^1^H 16 channel foot/ankle coil (Foot/Ankle 16 Coil, Siemens Healthineers, Erlangen, Germany).

Similar to previous studies values for aTSC were calculated from the acquired ^23^Na images by using reference phantoms (diameter 1 cm and height 3.5 cm) with 4% agarose content by weight (ROTI^®^Garose, Carl ROTH GmbH & Co. KG, Karlsruhe, Germany) [31,32]. In total, four reference phantoms were placed behind the dual-tuned coil with the different ^23^Na concentrations of 50 mM, 75 mM, 100 mM and 125 mM.

#### 4.2.1. ^23^Na Coil Sensitivity Correction

In line with previous studies, to correct for the spatial dependent sensitivity of the ^23^Na coil, first, homogenous water phantoms with ^23^Na concentrations of 154 mM were measured [25,31,32]. One cylindrical phantom (diameter 18 cm and height 11 cm) was placed in front of the coil, where the ATs of the participants were later placed. Two cuboid phantoms (23 cm length, 13 cm width and 6 cm height) were placed to the left and right behind the coil, because at the very center behind the coil a plastic shroud protects the cabling of the coil and hinders the stable positioning of one large phantom behind the coil. The sensitivity behind the coil was also corrected, because the reference phantoms for ^23^Na concentration estimation were later placed behind the coil to the left and right of aforementioned protective shroud. Imaging was conducted with the DA-3D-RAD sequence. The excitement pulse duration was set to 0.5 ms and the echo time (TE) to 0.3 ms. Twenty averages were measured to reduce noise. The remaining imaging parameters are documented in Table 4.

#### 4.2.2. Protocol 1: ^23^Na T_1_ Relaxation Times

For anatomical reference, ^1^H images were acquired using the dual-tuned coil and the DA-3D-RAD sequence. For determining ^23^Na T_1_ relaxation times in the AT, ^23^Na images were acquired with five different repetition times (TR) in a relatively high isotropic resolution of 2 mm × 2 mm × 2 mm to reduce the influence of the proximal skin tissue over partial volume effects as much as possible. A very short TE of 0.1 ms was used to maximize the sodium signal of the tendon. The excitation pulse duration was reduced to 0.16 ms to be able to measure images with TE = 0.1 ms. The measured TRs were kept relatively short to keep the total examination time reasonable, resulting in an examination time of 55:50 min:s with a maximum TR of 25 ms. Additional imaging parameters are documented in Table 4.

#### 4.2.3. Protocol 2: ^23^Na T_2_* Relaxation Times

Again, ^1^H images were acquired as an anatomical reference with the dual-tuned coil and the DA-3D-RAD sequence. Twelve different TEs were used to acquire ^23^Na images for determining ^23^Na T_2_* relaxation times in the AT. For this purpose, the DA-3D-RAD sequence was used in multi-echo mode three separate times with four different TEs each time and acquired in an interleaved pattern. Because of the readout time of 5 ms, which limits the minimal spacing between TEs per acquisition, it is not possible to measure more TEs in one run of the DA-3D-RAD sequence in a meaningful interval. The acquired projections were lowered to 40,000 to reduce measurement time, leading to an examination time of 01:00:00 h:min:s. The remaining imaging parameters are referenced in Table 4.

#### 4.2.4. Protocol 3: ^1^H T_2_* Relaxation Times, aTSC and Clinical Imaging

The dual-tuned surface coil was used to determine ^1^H T_2_* relaxation times in the ATs of participants. ^1^H images were acquired with the DA-3D-RAD sequence in multi-echo mode. The following TEs were measured: 0.1 ms, 3 ms, 6 ms and 9 ms. Afterwards the dual-tuned coil was used with the DA-3D-RAD sequence to acquire ^23^Na images. Further imaging parameters are shown in Table 4.

After imaging with the dual-tuned coil was completed, the coils were switched to the ^1^H 16 channel foot/ankle coil and the participants were repositioned. The ankle was imaged with proton-density (PD) weighted sequences in sagittal, transversal and coronal direction and a T_1_ weighted sequence in sagittal direction. Further parameters for the imaging sequences used in combination with the foot/ankle coil are listed in Table 5.

### 4.3. Image Post-Processing

ROIs containing the AT were drawn by M.F. on the ^1^H DA-3D-RAD images. The images were loaded into the software ITK-SNAP (v3.8.0, Cognitica, Philadelphia, PA, USA) and the contour of the AT was delineated manually using the images in transversal direction [56]. The ROI of the AT was divided into three parts along the transversal axis. The first 3 cm (measured as 30 ^1^H slices) beginning at the calcaneal insertion of the AT were labelled INS, the following 3 cm were labelled according to the middle portion of the tendon MID, and from there on the next 3 cm were labelled as the myotendinous junction MTJ [21].

ROIs were manually defined for each of the four reference phantoms behind the coil and a dedicated “noise” ROI was defined in a non-signal area but within the sensitivity profile of the coil for SNR calculation. These ROIs were used for ^23^Na parameter calculations after just interpolating them onto the differing resolution of the ^23^Na images, because the ^1^H and ^23^Na images were both acquired with the DA-3D-RAD sequence in the same position, orientation and FOV without repositioning of the patient.

All images acquired with the DA-3D-RAD sequence were reconstructed using a Hann Filter to reduce Gibbs ringing and increase SNR. The ^23^Na images for relaxation time calculation were motion-corrected due to the long acquisition times and the consequent expected movements of the participants. The in-house developed software stroketool was used, utilizing a cross-correlation algorithm based on advanced normalization tools for image registration [57,58]. The ^23^Na relaxation times and aTSCs were calculated with in-house developed MATLAB (MathWorks, Natick, MA, USA, R2018a) scripts. The average values in the corresponding ROIs were used to achieve more stable results than would be the case with voxel-wise fitting.

For the determination of ^23^Na T_1_ the data of protocol 1 was fitted according to the following relation for the signal S(TR):(1)$$S\left(\mathrm{TR}\right)=S_{0}\cdot \left(1-e^{\frac{-\mathrm{TR}}{T_{1}}}\right)+\mathrm{noise}$$

The ^23^Na T_2_* relaxation times were estimated fitting the data of protocol 2 biexponentially, since ^23^Na has a nuclear spin of 3/2 and therefore has a short (T_2s_*) and a long (T_2l_*) transversal relaxation component in the AT [27]. The data were fitted according to the following relation for the signal S(TE):(2)$$S\left(\mathrm{TE}\right)=S_{0}\cdot \left(p_{s}\cdot e^{\frac{-\mathrm{TE}}{T_{2s}^{*}}}+\left(1-p_{s}\right)\cdot e^{\frac{-\mathrm{TE}}{T_{2l}^{*}}}\right)+\mathrm{noise}$$

Here p_s_ denotes the fraction of the short relaxation component T_2s_* in the transversal relaxation that satisfies the condition 0 < p_s_ < 1.

For ^1^H T_2_* relaxation time calculation MATLAB was used to generate ^1^H T_2_* relaxations time maps based on the ^1^H data of the DA-3D-RAD sequence. These relaxation times were calculated voxel-wise with a monoexponential fitting algorithm and the offset as a free parameter. The ^23^Na SNR was calculated by dividing ^23^Na signal in the AT ROIs by the standard deviation of the ^23^Na signal in the noise ROI for each participant measured with protocol 3.

The ^23^Na signal of the agarose reference phantoms was linearly fitted according to their concentrations and the aTSCs in the ATs of the participants were calculated based on that linear fit. The aTSCs were corrected based on the difference in ^23^Na relaxation times between the reference phantoms and the ATs, which would otherwise influence their signal ratio based on the chosen TR and TE values. For this purpose, the average ^23^Na relaxation times across all participants were used for the INS, MID and MTJ portions of the tendons and the average of the ^23^Na relaxation times of the reference phantoms was calculated across participant measurements of protocol 1 and 2 and different ^23^Na concentrations in the phantoms.

In addition, a partial volume correction was applied, which was previously published by our group [31,32]. This correction is necessary, because the size of the AT in sagittal and coronal direction is comparable to the 2 mm voxel edge size in the ^23^Na images, leading to an underestimation of the ^23^Na signal in the tendon. For counteracting this effect, after interpolating the ^1^H ROI to ^23^Na resolution size, the number of higher resolution ^1^H tendon voxels were counted in each lower resolution ^23^Na voxel. This volume fraction of tissue contributing to the ^23^Na signal was averaged over all ^23^Na voxels in the tendon ROI and the inverse of this fraction was multiplied by the aTSC of the tendon. In this way, the influence of averaging the tendon ^23^Na signal over the fractions of tissues that do not contribute to the signal was reduced.

### 4.4. Statistical Analysis

Statistical analysis was conducted using SPSS (IBM Corp. Released 2020. IBM SPSS Statistics for Windows, Version 27.0. Armonk, NY, USA: IBM Corp.). Descriptive statistics (mean, standard deviation) were performed for all established ^1^H and ^23^Na parameters based on measurements in healthy controls. For the patient the mean and standard deviation values of aTSC, ^23^Na SNR and ^1^H T_2_* were calculated based on the different values of the voxels in the AT ROIs. The results of aTSC, ^23^Na SNR, ^1^H T_2_* and their respective standard deviations in % in INS, MID and MTJ were tested on significant difference using the Friedman ANOVA with a significance level of p ≤ 0.05 for the ROI values of the healthy controls. When *p* of the Friedman ANOVA was below 0.05, the three groups INS, MID and MTJ were further tested against each other using three Wilcoxon rank-sum tests (INS-MID, INS-MTJ, MID-MTJ) and the *p*-values of these rank-sum tests were multiplied by three to correct according to Bonferroni [59].

## 5. Conclusions

For the first time, aTSC and ^23^Na relaxation times were successfully measured in the ATs of a healthy control cohort and one pilot patient using sodium MRI. Previously established parameters for biochemical examination of the AT, namely ^23^Na SNR and ^1^H T_2_* mapping, have also been determined and compared to aTSC. Significant regional differences across the ATs of healthy controls have been observed and may be linked to the different loading scenarios of each part of the tendon resulting in different biochemical compositions. In conclusion, our study could provide a foundation on which future studies investigating ATs could be built and compared more reliably.

## Figures and Tables

**Figure 1 ijms-23-10890-f001:**
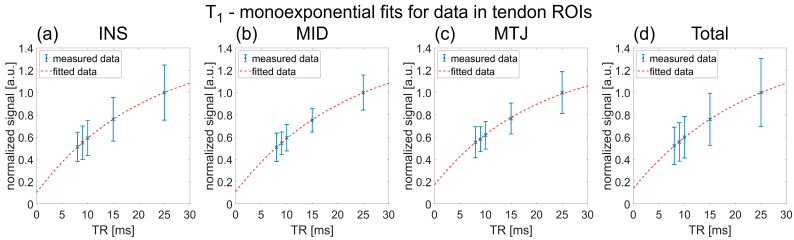
Data points and fitting results of ^23^Na T_1_ relaxation times in the different Achilles tendon (AT) regions of interest (ROIs) of an exemplary healthy control. ROIs were named after their position, where (**a**) corresponded to the most distal 3 cm of the AT, i.e., the distal 3 cm of the tendon’s length extending into the calcaneal insertion (INS), (**b**) corresponded to the middle 3 cm, i.e., the middle portion of the AT (MID), and (**c**) corresponded to the proximal 3 cm, which was referred to as the myotendinous junction (MTJ). In (**d**), the data and fitting of the total ROI combining all three ROIs, i.e., INS, MID and MTJ are displayed. The corresponding parameter results were (**a**) ^23^Na T_1_ = 21.2 ms, *R*^2^ = 0.999, (**b**) ^23^Na T_1_ = 22.1 ms, *R*^2^ = 0.999, (**c**) ^23^Na T_1_ = 19.6 ms, *R*^2^ = 0.997 and (**d**) ^23^Na T_1_ = 23.1 ms, *R*^2^ = 0.999.

**Figure 2 ijms-23-10890-f002:**
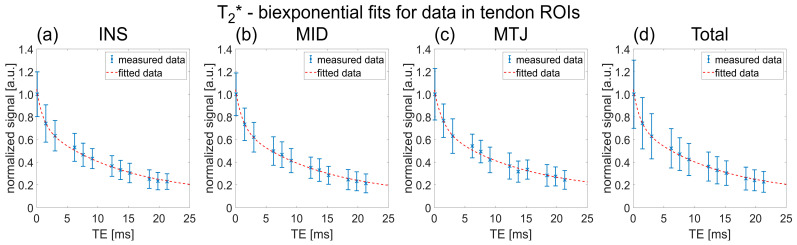
Data points and fitting results to determine ^23^Na T_2_* relaxation times in the different AT ROIs of an exemplary healthy control. The corresponding parameter results were (**a**) ^23^Na T_2s_* = 1.0 ms, ^23^Na T_2l_* = 13.3 ms, p_s_ = 32.7%, R^2^ = 0.997, (**b**) ^23^Na T_2s_* = 1.2 ms, ^23^Na T_2l_* = 13.6 ms, p_s_ = 34.9%, R^2^ = 0.998, (**c**) ^23^Na T_2s_* = 1.4 ms, ^23^Na T_2l_* = 13.8 ms, p_s_ = 35.7%, R^2^ = 0.994 and (**d**) ^23^Na T_2s_* = 1.1 ms, ^23^Na T_2l_* = 13.3 ms, p_s_ = 33.2%, R^2^ = 0.998.

**Figure 3 ijms-23-10890-f003:**
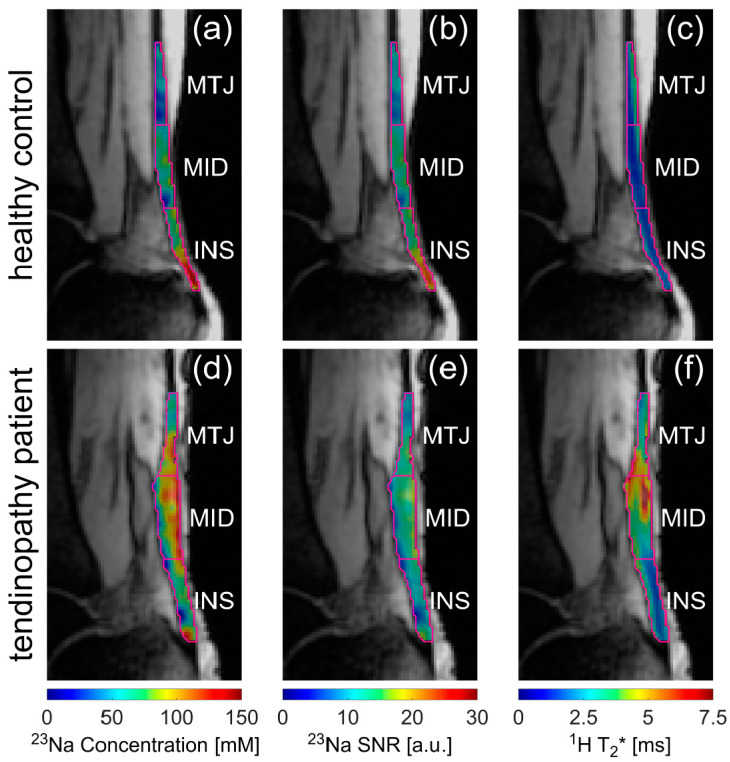
Apparent tissue sodium concentration (aTSC) (**a**,**d**) and ^23^Na signal-to-noise ratio (SNR) (**b**,**e**) as well as ^1^H T_2_* maps (**c**,**f**) overlaid onto the ^1^H images acquired with the density-adapted radial sequence using TE = 9 ms. The overlaid colormaps are displayed for an exemplary healthy control (**a**–**c**) and the patient with Achilles tendinopathy (**d**–**f**). The different AT ROIs (INS, MID and MTJ) are outlined and labelled according to their position.

**Figure 4 ijms-23-10890-f004:**
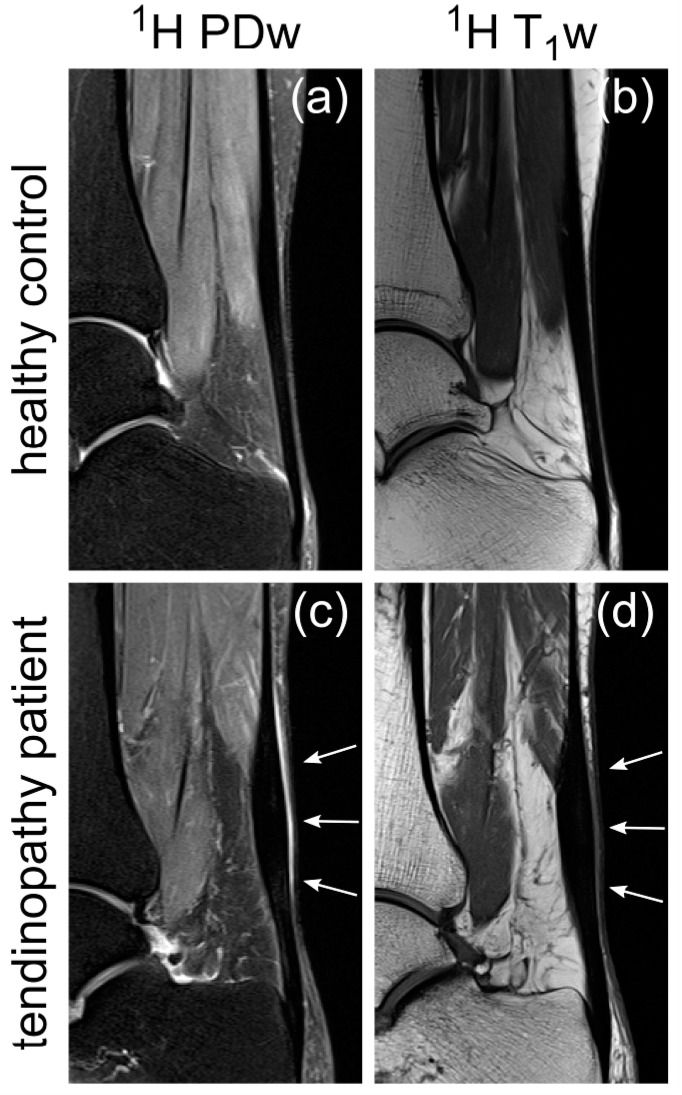
Sagittal proton density (PD) (**a**,**c**) and T_1_ (**b**,**d**) weighted images of an exemplary healthy control (**a**,**b**) and the patient with Achilles tendinopathy (**c**,**d**) acquired as anatomical reference in line with clinical standard procedures. The patient showed local thickening mainly in the middle part of the AT with adjacent peritendinous fluid in the same region, which is both highlighted in (**c**,**d**) with white arrows.

**Figure 5 ijms-23-10890-f005:**
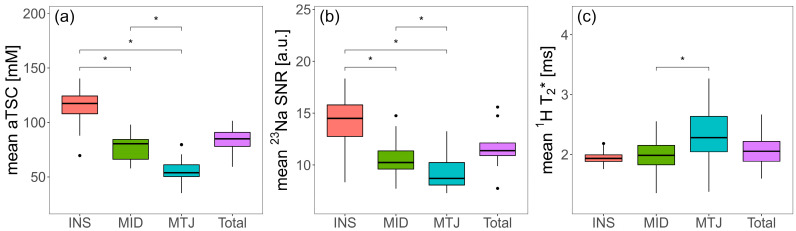
Boxplots illustrating the results for healthy controls for (**a**) aTSC, (**b**) ^23^Na SNR and (**c**) ^1^H T_2_* for the AT ROIs INS, MID, MTJ and the total ROI, which is the combination of the previous three. The parentheses above the boxplots indicate whether the difference between the groups was significant (*) in the Bonferroni corrected Wilcoxon rank-sum tests with a significance level of *p* ≤ 0.05.

**Table 1 ijms-23-10890-t001:** Fitting results for ^23^Na relaxation times measured in three healthy controls in the different Achilles tendon (AT) regions of interests (ROIs) and 4% agarose phantoms with different ^23^Na concentrations.

ROI	^23^Na T_1_ [ms]	*R*^2^ (^23^Na T_1_ Fitting)	^23^Na T_2s_* [ms]	^23^Na T_2l_* [ms]	p_s_ [%]	*R*^2^ (^23^Na T_2_* Fitting)
INS	18.4 ± 2.7	0.994 ± 0.007	1.4 ± 0.4	14.5 ± 1.4	30.4 ± 2.7	0.994 ± 0.003
MID	19.2 ± 2.5	0.999 ± 0.001	1.4 ± 0.3	14.2 ± 1.0	32.6 ± 2.7	0.995 ± 0.003
MTJ	23.3 ± 7.2	0.996 ± 0.003	1.5 ± 0.5	14.6 ± 0.8	31.8 ± 3.6	0.986 ± 0.011
Total	20.4 ± 2.4	0.998 ± 0.002	1.4 ± 0.4	13.9 ± 0.8	31.6 ± 2.6	0.995 ± 0.004
125 mM Phantom	40.5 ± 2.6	0.998 ± 0.002	5.8 ± 0.3	14.6 ± 0.7	52.8 ± 0.8	0.997 ± 0.001
100 mM Phantom	40.0 ± 4.9	0.996 ± 0.001	5.7 ± 0.5	14.2 ± 0.6	49.9 ± 4.7	0.993 ± 0.009
75 mM Phantom	37.5 ± 2.2	0.993 ± 0.004	6.6 ± 0.1	11.8 ± 0.2	67.7 ± 1.3	0.987 ± 0.010
50 mM Phantom	35.9 ± 4.6	0.979 ± 0.012	5.8 ± 0.7	11.2 ± 0.3	70.6 ± 0.9	0.991 ± 0.001
Phantom mean	38.5 ± 3.8	0.991 ± 0.009	6.0 ± 0.5	13.0 ± 1.6	60.2 ± 9.6	0.992 ± 0.007

Abbreviations: ROI—region of interest, INS—tendon insertion point into the calcaneus, MID—middle portion of the tendon, MTJ—myotendinous junction, p_s_—fraction of T_2s_* of the total T_2_* relaxation, *R*^2^—coefficient of determination for fits.

**Table 2 ijms-23-10890-t002:** Parameter results for aTSC, ^23^Na SNR and ^1^H T_2_* in all ten healthy controls and the patient with Achilles tendinopathy in the different AT ROIs. Two different standard deviations were calculated: While “±” indicates the absolute deviation between the mean values of the controls, the column “SD” displays the deviation between voxel values in the ROI of each control in percent.

Parameter	ROI	Controls		Patient	
		Mean ± SD	SD [%]	Mean ± SD	SD [%]
aTSC [mM]	INS	112.9 ± 21.1	28.6 ± 4.4	89.3 ± 37.7	42.2
	MID	77.3 ± 13.3	26.1 ± 3.9	90.6 ± 35.2	38.8
	MTJ	55.3 ± 13.3	28.0 ± 5.5	51.3 ± 19.2	37.4
	Total	82.2 ± 13.9	36.6 ± 8.0	76.5 ± 33.1	43.3
^23^Na SNR [a.u.]	INS	14.2 ± 2.8	24.3 ± 3.1	11.8 ± 3.7	31.2
	MID	10.7 ± 2.2	20.7 ± 2.0	12.3 ± 3.5	28.8
	MTJ	9.5 ± 2.1	20.2 ± 1.6	9.5 ± 2.2	23.2
	Total	11.7 ± 2.2	29.4 ± 5.4	11.3 ± 3.5	30.7
^1^H T_2_* [ms]	INS	1.9 ± 0.1	25.5 ± 5.3	2.7 ± 0.8	30.6
	MID	2.0 ± 0.3	32.9 ± 3.5	3.7 ± 1.3	34.4
	MTJ	2.3 ± 0.6	33.1 ± 5.3	3.5 ± 1.0	28.1
	Total	2.1 ± 0.3	32.9 ± 2.7	3.3 ± 1.1	34.3

Abbreviations: SD—Standard deviation of relaxation times within a region, aTSC—apparent tissue sodium concentration, SNR—signal-to-noise ratio.

**Table 3 ijms-23-10890-t003:** Resulting *p*-values for testing significant differences between the different ROIs in the AT for the healthy controls. The parameters aTSC, ^23^Na SNR and ^1^H T_2_* and their relative standard deviations were tested with a significance level of *p* ≤ 0.05.

Parameters Tested	Friedman-ANOVA *p*-Values	Wilcoxon Rank-Sum Test *p*-Values		
		INS-MID	INS-MTJ	MID-MTJ
mean aTSC	**<0.001**	**0.015**	**0.015**	**0.015**
SD aTSC	0.202	-	-	-
mean ^23^Na SNR	**<0.001**	**0.015**	**0.015**	**0.015**
SD ^23^Na SNR	**0.008**	0.065	**0.028**	1.000
mean ^1^H T_2_*	**0.020**	0.854	0.178	**0.038**
SD ^1^H T_2_*	0.061	-	-	-

**Table 4 ijms-23-10890-t004:** Parameters of the DA-3D-RAD sequence used for imaging with the dual-tuned ^23^Na/^1^H surface coil.

	^23^Na Coil	Protocol 1	Protocol 2	Protocol 3	^1^H Imaging
	Sensitivity	(^23^Na T_1_)	(^23^Na T_2_*)	(aTSC)	
Sequence type	DA-3D-RAD	DA-3D-RAD	DA-3D-RAD	DA-3D-RAD	DA-3D-RAD
Nucleus	^23^Na	^23^Na	^23^Na	^23^Na	^1^H
Orientation	sag	sag	sag	sag	sag
Repetition time [ms]	15	8/9/10 /15/25	30	15	12
Echo time [ms]	0.3	0.1	[0.1/6.2/12.3/18.4] [1.5/7.6/13.7/19.8] [3.0/9.1/15.2/21.3]	0.1	0.1/3.0 /6.0/9.0
Field of View [mm^3^]	180 × 180 × 180	180 × 180 × 180	180 × 180 × 180	180 × 180 × 180	180 × 180 × 180
Number of Projections	50,000	50,000	40,000	50,000	25,000
Voxel size [mm^3^]	2 × 2 × 2	2 × 2 × 2	2 × 2 × 2	2 × 2 × 2	1 × 1 × 1
Flip angle [°]	90	90	90	90	5
Pulse duration [ms]	0.5	0.16	0.16	0.16	0.16
Readout time [ms]	5	5	5	5	1
Signal averages	20	1	1	1	1
Total examination time [h:min:s]	04:10:00	00:55:50	01:00:00	00:12:30	00:05:00

Abbreviations: sag—sagittal, DA-3D-RAD—density-adapted 3D radial.

**Table 5 ijms-23-10890-t005:** Imaging sequences and their parameters used in combination with the ^1^H foot/ankle coil for clinical standard examination.

	PD-Weighted fs	PD-Weighted fs	PD-Weighted fs	T_1_-Weighted
Sequence type	TSE	TSE	TSE	TSE
Turbo Factor	9	9	9	2
Grappa	2	2	2	2
Orientation	sag	tra	cor	sag
Repetition time [ms]	3150	3940	3290	805
Echo time [ms]	42	42	44	14
Field of View [mm]	280 × 280	280 × 280	180 × 180	280 × 280
Image matrix [px]	704 × 704	640 × 640	512 × 512	832 × 832
Pixel size [mm]	0.40 × 0.40	0.44 × 0.44	0.35 × 0.35	0.34 × 0.34
Flip angle [°]	150	150	150	140
Slices	20	56	40	20
Slice gap [mm]	0.3	0.3	0.3	0.3
Slice thickness [mm]	3	3	3	3
Examination time [min:s]	02:03	03:49	02:45	02:55

Abbreviations: tra—transversal, cor—coronal, PD—proton density, fs—fat saturated, TSE—turbospin-echo, GRAPPA—generalized autocalibrating partial parallel acquisition.

## Data Availability

Data can be provided by the authors upon reasonable request.

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
