# Peer review of "Evaluation of Sodium Relaxation Times and Concentrations in the Achilles Tendon Using MRI"

_ijms, 2022, doi:10.3390/ijms231810890_

Round 1

Reviewer 1 Report

The authors reported a novel method to accurately measure the aTSC and 23Na relaxation times in the AT. They also evaluated this method against other previously established procedures, and further hypothesized it's possible to provide foundations for AT investigation. This whole paper is easy to read and nice written. The experiment is well designed and the conclusion is solid and sound. I do have one concern on the sodium concentration experiment. In the paper, the authors studied the sodium concentration range from 0- 150 mM. I was wondering how representative this range is. Could the authors give any explanations? Therefore I recommend this paper to be accepted after minor revision.  

Author Response

Answer to reviewer 1:

General Comments

Point 1: The authors reported a novel method to accurately measure the aTSC and 23Na relaxation times in the AT. They also evaluated this method against other previously established procedures, and further hypothesized it's possible to provide foundations for AT investigation. This whole paper is easy to read and nice written. The experiment is well designed and the conclusion is solid and sound.

Author’s response: We thank the reviewer for his / her thorough Review of our manuscript and the conveyed overall appreciation.

Point 2: I do have one concern on the sodium concentration experiment. In the paper, the authors studied the sodium concentration range from 0- 150 mM. I was wondering how representative this range is. Could the authors give any explanations? Therefore I recommend this paper to be accepted after minor revision.

Author’s response: We thank the reviewer for this valuable comment. We are not aware of any study in which the exact sodium concentration in the Achilles tendon has been measured, either by MRI or by other methods such as histologic analysis. 140 to 150 mM are often named as reference for physiological in-vivo sodium concentrations, because that is the approximate range of sodium concentration in blood serum measured by lab analysis and the sodium concentration in the extracellular space. The Achilles tendon is mainly composed of a collagen extracellular matrix, which would indicate the Achilles tendon should have a sodium concentration closer to 140 mM [1,2].

However, it has been previously reported that sodium concentrations measured by MRI are lower than those determined by chemical analysis, which means that our values probably also slightly underestimate the actual sodium concentrations in the Achilles tendon [3]. The reasons for this could be a stronger influence of partial volume effects than expected from our correction method or spatial B0 and B1 field variations between the regions of the agarose reference phantoms and the Achilles tendon. This is also one of the reasons why we refer to our values for sodium concentration as "apparent tissue sodium concentration" (aTSC), because our values still need to be validated by, for example, chemical analyses in perhaps future studies.

Author’s action: To clarify the issue, an additional explanation has been inserted in the discussion from line 237 to line 249: “Our result of approximately 82 mM for the aTSC in the whole AT is considerably lower than the expected sodium concentration of 140 mM in the extracellular space [39]. The AT is mainly composed of a collagen-rich extracellular matrix, which would indicate the aTSC values should be closer to the extracellular sodium concentration of 140 mM [40]. However, sodium concentration measurements using MRI have previously been reported to obtain lower values than, for example, chemical analysis, which might also be the case for our aTSC values for the AT [41]. Reasons could be an underestimation of the influence of partial volume effects in our correction or spatial deviations of the B0 and B1 fields between the positions of the agarose reference phantoms and the AT. This is another reason why our resulting sodium concentrations should be referred to as "apparent tissue sodium concentration", because validation of the accuracy of measurement of sodium concentration in the AT by MRI would need to be investigated in a future study by comparison with, for example, chemical analysis.”

  1. Freedman, B.R. The Achilles Tendon: Fundamental Properties and Mechanisms Governing Healing. Muscles. Ligaments Tendons J. 2014, 4 (2), 245–255. https://doi.org/10.11138/mltj/2014.4.2.245.
  2. Madelin, G.; Kline, R.; Walvick, R.; Regatte, R.R. A Method for Estimating Intracellular Sodium Concentration and Extracellular Volume Fraction in Brain in Vivo Using Sodium Magnetic Resonance Imaging. Sci. Rep. 2014, 4, 1–7. https://doi.org/10.1038/srep04763.
  3. Kopp, C.; Linz, P.; Wachsmuth, L.; Dahlmann, A.; Horbach, T.; Schöfl, C.; Renz, W.; Santoro, D.; Niendorf, T.; Müller, D.N.; et al. 23 Na Magnetic Resonance Imaging of Tissue Sodium. Hypertension 2012, 59 (1), 167–172. https://doi.org/10.1161/HYPERTENSIONAHA.111.183517.

Reviewer 2 Report

A generally well written manuscript on sodium MRI for AT.
The presentation, can however be improved.   
For the title, suggest to include a word on MRI to reflect the topics in discussion.

Minor suggestion

1. Page 1, Line 10, do not begin a sentence with the word (To ...), and correct this throughout the whole manuscript.
2. Page 2, this sentence is incomplete and sounds weird
We aim to determine 23Na parameters for the AT, measuring 23Na T1 and T2* relaxa- 87
tion times and applying the aforementioned correction methods using the calculated re- 88
laxation times to estimate aTSC with a clinical 3 Tesla (T) MR scanner.

2. Fig 3 (superscript and subscript usage) 23Na, 1H and T2, and the rest of the figures in this manuscript.
3. Page 7, To the best of our knowledge, ...
4. How many actual patients were used, and explain why or if this is enough ?

Author Response

Answer to reviewer 2:

General Comments

Point 1: A generally well written manuscript on sodium MRI for AT. The presentation, can however be improved.  

Author’s response: We thank the reviewer for his / her thorough Review of our manuscript and the conveyed overall appreciation.

Point 2: For the title, suggest to include a word on MRI to reflect the topics in discussion.

Author’s response and action: We agree with the reviewer and modified the title of the article to specify the study was conducted using MRI: “Evaluation of sodium relaxation times and concentrations in the Achilles tendon using MRI”.

Specific Comments

Point 3: Page 1, Line 10, do not begin a sentence with the word (To ...), and correct this throughout the whole manuscript.

Author’s response: We agree with the reviewer and changed the sentences according to his / her suggestion.

Author’s action: We have revised all sentences so that they do not begin with "to". Below we state the line and the new wording of the sentences.

Line 29 - 30: “23Na T1 and T2* of the AT were also measured for three controls to correct for different relaxation behavior.”

Line 32 - 34: “These are the first reported aTSCs and 23Na relaxation times for the AT using sodium MRI and may serve for future comparability in different studies regarding examinations of diseased ATs with sodium MRI.”

Line 60 - 61: “A variety of different quantitative MRI techniques have been proposed to detect these biochemical changes.”

Line 64 - 66: “Other techniques like chemical exchange saturation transfer (CEST), T1ρ and 23Na imaging indicate promising results to assess biochemical components like PG and GAG content [17–20].”

Line 199 - 201: “Values for 23Na relaxation times in the AT using MRI were not previously published to the best of our knowledge, which inhibits the direct comparison of our values to literature.”

Line 229 - 231: “There is no published study investigating aTSC in ATs or at least in other tendons or ligaments to the best of our knowledge, so a comparison of aTSC with literature values is difficult.”

Line 305 - 309: “Other approaches towards accelerating 23Na imaging would have to be used in conjunction with more projections and signal averages to obtain more 23Na signal in the tendon. This would allow to further increase imaging resolution while keeping examination duration reasonable.”

Line 352 - 355: “Similar to previous studies values for aTSC were calculated from the acquired 23Na images by using reference phantoms (diameter 1 cm and height 3.5 cm) with 4 % agarose content by weight (ROTI®Garose, Carl ROTH GmbH & Co. KG, Karlsruhe, Germany) [31,32].”

Line 379 - 380: “The excitation pulse duration was reduced to 0.16 ms to be able to measure images with TE = 0.1 ms.”

Line 380 - 382: “The measured TRs were kept relatively short to keep the total examination time reasonable, resulting in an examination time of 55:50 min:s with a maximum TR of 25 ms.”

Line 386 - 387: “Twelve different TEs were used to acquire 23Na images for determining 23Na T2* relaxation times in the AT.”

Line 391 - 393: “The acquired projections were lowered to 40000 to reduce measurement time, leading to an examination time of 01:00:00 h:min:s.”

Line 395 - 397: “The dual-tuned surface coil was used to determine 1H T2* relaxation times in the ATs of participants. 1H images were acquired with the DA-3D-RAD sequence in multi-echo mode.”

Line 426 - 427: “All images acquired with the DA-3D-RAD sequence were reconstructed using a Hann Filter to reduce Gibbs ringing and increase SNR.”

Line 429 - 431: “The in-house developed software stroketool was used, utilizing a cross-correlation algorithm based on advanced normalization tools for image registration [57,58].”

Line 463 - 465: “For counteracting this effect, after interpolating the 1H ROI to 23Na resolution size, the number of higher resolution 1H tendon voxels were counted in each lower resolution 23Na voxel.”

Point 4: Page 2, this sentence is incomplete and sounds weird: We aim to determine 23Na parameters for the AT, measuring 23Na T1 and T2* relaxation times and applying the aforementioned correction methods using the calculated relaxation times to estimate aTSC with a clinical 3 Tesla (T) MR scanner.

Author’s response: We agree with the reviewer and thank her / him for the suggestion.

Author’s action: We changed the wording of the sentence in Line 86 - 88: “We aim to determine 23Na parameters for the AT using a clinical 3 Tesla (T) MR scanner by first measuring 23Na T1 and T2* relaxation times. Then we use the calculated relaxation times and apply the correction methods mentioned above to determine aTSC.”

Point 5: Fig 3 (superscript and subscript usage) 23Na, 1H and T2, and the rest of the figures in this manuscript.

Author’s response and action: We thank the reviewer for pointing out this issue and changed all figures in the manuscript to have appropriate superscript and subscript usage.

Point 6: Page 7, To the best of our knowledge, ...

Author’s response and action: We agree with the reviewer and changed the text according to his / her suggestion.

Line 199 - 200: “Values for 23Na relaxation times in the AT using MRI were not previously published to the best of our knowledge,…“

Line 229 - 230: “There is no published study investigating aTSC in ATs or at least in other tendons or ligaments to the best of our knowledge,…”

Point 7: How many actual patients were used, and explain why or if this is enough?

Author’s response: We thank the reviewer for bringing attention to this important point. The number of actual patients measured were only one, as is stated in Materials and Methods in the lines 327 - 329: “Ten healthy controls (six females, four males, mean age 25.4 ± 0.9 years) and one patient with tendinopathy of the AT (female, 55 years, established by patient history and clinical MRI) participated in this study.” While the manuscript already contains a short paragraph about the resulting limited capability of our data to compare values between healthy controls and patients in the Discussion in lines 313 - 319, we agree it should be stated more clearly that we only measured one patient.

Author’s action: In the Discussion we changed line 313 - 315: “Our results have limited capability for comparing the 23Na values of the healthy control groups with patients, because we only measured one patient. For a reliable comparison, a much higher number of patients would have to be measured in a future study.”